# Diagnosis of Tuberculosis in a Case of Chronic Urticaria following Omalizumab Therapy

**DOI:** 10.3390/antibiotics12121655

**Published:** 2023-11-24

**Authors:** Alberto Zolezzi, Gina Gualano, Maria A. Licata, Silvia Mosti, Paola Mencarini, Roberta Papagni, Antonella Vulcano, Angela Cannas, Alberta Villanacci, Fabrizio Albarello, Franca Del Nonno, Daniele Colombo, Fabrizio Palmieri

**Affiliations:** 1UOC Malattie Infettive dell’Apparato Respiratorio, Istituto Nazionale per le Malattie Infettive “L. Spallanzani” IRCCS, 00149 Roma, Italy; gina.gualano@inmi.it (G.G.); silvia.mosti@inmi.it (S.M.); paola.mencarini@inmi.it (P.M.); fabrizio.palmieri@inmi.it (F.P.); 2UOC Laboratorio di Microbiologia, Istituto Nazionale per le Malattie Infettive “L. Spallanzani” IRCCS, 00149 Roma, Italy; 3UOSD Diagnostica per Immagini delle Malattie Infettive, Istituto Nazionale per le Malattie Infettive “L. Spallanzani” IRCCS, 00149 Roma, Italy; alberta.villanacci@inmi.it (A.V.);; 4UOSD Istologia, Citologia e Anatomia Patologica, Istituto Nazionale per le Malattie Infettive “L. Spallanzani” IRCCS, 00149 Roma, Italy; franca.delnonno@inmi.it (F.D.N.); daniele.colombo@inmi.it (D.C.)

**Keywords:** tuberculosis, screening, omalizumab, biotechnological treatments

## Abstract

In Italy, tuberculosis (TB) incidence in the last decade has remained constant at under 10 cases/100,000 inhabitants. In the Philippines, TB annual incidence is greater than 500 cases/100,000 inhabitants. Omalizumab is a humanized anti-IgE monoclonal antibody approved for the treatment of chronic spontaneous urticaria. We report the case of a 32-year-old Filipino woman who suffered from chronic urticaria, treated with topic steroids since June 2022 and systemic steroids for 2 weeks. In November 2022, she started omalizumab therapy at a monthly dose of 300 mg; she was not screened for TB infection. In the same month, a left laterocervical lymphadenopathy arose, which worsened in February 2023 (diameter: 3 cm). The patient recovered in April 2023 in INMI “Lazzaro Spallanzani” in Rome for suspected TB. Chest CT showed a “tree in bud” pattern at the upper-right pulmonary lobe. The patient tested positive for lymph node biopsy molecular tuberculosis. The patient started standard antituberculosis therapy. She discontinued omalizumab. To our knowledge, this is the second diagnosed TB case during omalizumab treatment, which suggests that attention should be paid to the known risk of TB during biotechnological treatments. Even if current guidelines do not recommend screening for TB before starting anti-IgE therapy, further data should be sought to assess the relationship between omalizumab treatment and active TB. Our experience suggests that screening for TB should be carried out in patients from highly tuberculosis-endemic countries before starting omalizumab therapy.

## 1. Introduction

Tuberculosis (TB) is a chronic infectious disease that has a massive global burden. It is caused by Mycobacterium (M.) tuberculosis [1].

In Italy, TB is a relatively rare disease, with an incidence lower than 10 cases/100,000 inhabitants in last 10 years. Incidence in Italy decreased from 6 cases/100,000 inhabitants in 2017 to 4 cases/100,000 inhabitants in 2021 [2]. The Philippines is among the most TB-endemic countries, with TB annual incidence greater than 500 cases/100,000 inhabitants [1].

Humans are the only important reservoirs of infection for *Mycobacterium tuberculosis* and it has been historically estimated that as many as one third of the world’s population harbors TB infection. *M. tuberculosis* infection can be screened using either a tuberculin skin test (TST) or an interferon gamma release assay (IGRA). The lifetime risk of developing active tuberculosis has generally been held to be in the range of 5–10% of TB-infected people [3], and the treatment of TB-infected persons with isoniazid for 9 months has been found to reduce the risk of progression to active tuberculosis by about 80% [4,5].

The most common site of extrapulmonary TB consists of lymphatic involvement, followed by genitourinary, bone and joint, central nervous system, peritoneal, and other abdominal organ involvement [6].

Chronic spontaneous urticaria (CSU) is characterized by wheals, angioedema, or both that occur continuously or sporadically for at least 6 weeks. The wheals are intensely pruritic, resolve in <24 h, and tend to recur on a daily basis [7].

Omalizumab (Xolair^®^, is a registered trademark of Novartis AG, One Health Plaza, East Hanover, NJ 07936-1080) is a recombinant humanized monoclonal antibody against IgE that was approved by the Food and Drug Administration (FDA) for the treatment of antihistamine-refractory CSU. Omalizumab is also approved for high-dose steroid-refractory bronchial asthma and for steroid-refractory nasal rhinosinusitis with poliposis. It works by binding to free IgE and inhibiting interaction with the FcεRI receptor on basophils and mast cells, thus preventing their activation.

One meta-analysis, consisting of seven trials, reported that patients treated with omalizumab had significantly reduced itch scores and wheal scores compared to the placebo [8].

Omalizumab is generally well tolerated, and the most common adverse effects reported include injection site reactions, viral infections, upper respiratory infections, sinusitis, and headaches [9].

Only one single trial found a higher incidence of upper respiratory tract infection among patients with chronic idiopathic/spontaneous urticaria receiving omalizumab compared with the placebo (7.1% versus 2.4%, respectively) [10]. In addition, long-term post-marketing surveillance has shown that omalizumab therapy is not associated with an increased risk of other potentially immunosuppression-related adverse events, such as malignancy [11].

IgE is best known for its role in allergic disease. However IgE has been shown to play a role in antiviral immune responses to respiratory viruses, in offering protection from toxin exposure, as a sensor for small quantities, and as part of the immune response against parasites [12].

According to available guidelines, particular attention has been paid to the so-called geohelminths (or soil-transmitted helminths). For migrants coming from specific endemic areas, pretreatment screening may be extended to certain geographically restricted systemic helminth infections (i.e., filariasis or schistosomiasis), although evidence demonstrating increased omalizumab-induced susceptibility to these conditions is lacking [13].

A high risk of TB reactivation is linked to high-dose systemic steroid therapy that lasts for more than one month, as well as prednisolone at a dose of ≥15 mg daily (or its equivalent) for a duration extending beyond 1 month [14].

There are no studies supporting the correlation between omalizumab and tuberculosis. According to available data, only one patient reported in a prospective study had to discontinue omalizumab because of TB diagnosis [15], whereas another patient in a retrospective study had to discontinue omalizumab for tuberculosis, but he was in the control group [13].

In the same study, the adequate screening of at-risk subjects was proposed to avoid intestinal helminth infections, before the start of therapy [13]; the same method has not yet been adopted for TB risk. So, screening is no longer required for TB before omalizumab is administered for any approved indications of anti-IgE therapy [16].

We herein report a case of pulmonary and extrapulmonary TB during omalizumab administration for CSU in a patient who was not screened for TB before starting anti-IgE therapy.

## 2. Detailed Case Presentation

In April 2023, a 32-year-old Filipino woman in Italy was observed at our institution for lateral cervical lymphonodal swelling that had lasted for 8 years. She did not refer to any relevant pathology in remote pathologic history. The last time she travelled was to the Philippines in early 2022, where she stayed for 3 months. She was diagnosed with urticaria in June 2022 and was initially treated with topical therapy using oral antihistamine (desloratadine), before being treated with oral steroids (methylprednisolone 16 mg/die) and vitamins (diathynil) for a duration of two weeks.

For the persistence of urticaria, she was diagnosed with CSU in November 2022 and she started omalizumab therapy at a monthly dose of 300 mg. She was not tested for TB infection using either a tuberculin skin test (TST) or an interferon gamma release assay (IGRA). After a month of treatment, she underwent mild left non-painful laterocervical lymphadenopathy, which dramatically intensified in February 2023. Neck nuclear magnetic resonance (NMR) in March 2023 showed necrotic lymphadenopathy with a diameter of 3 cm, making contact with the omolateral internal jugular vein and internal carotid, which displaced the left submandibular gland. The patient did not experience any cough, fever, weight loss, or night sweating symptoms. She was referred to our institution in April 2023.

She never smoked and her profession was babysitting. She was not aware of any contact with tubercular patient.

A chest computed tomography (CT) examination was performed in our hospital, which showed a “Tree in bud” pattern at the upper-right pulmonary lobe (Figure 1); no pulmonary consolidation with air bronchogram was found. No pleural or pericardial effusion was found.

The patient tested negative for HIV, CMV, EBV, and toxoplasma and positive for IGRA (quantiferon-plus) following a lymphocitary stimulation test with antigen ESAT-6 and CFP-10 (CD4) (0.21 UI/mL), a lymphocitary stimulation test with antigen ESAT-6 and CFP (CD4 and CD8) (1.4 UI/mL), and immunocompetency control (9.87 UI/mL).

Blood tests did not show hypereosinophilia (7.800/μL of white blood cells, eosinophils 2.1%); her liver and renal function was normal after being adminstered with alanine amino transferase (ALT) (11 unity/milliliter), aspartate aminotransferase (AST) (5 unity/milliliter), creatinine (0.71 mg/dL), and IgE 558 (KU/I). The patient tested negaitive for mycobacteria and parasites (protozoa and helminths) in three fecal samples.

The objective skin examination showed facial hyperemia; the neck examination showed weekly painful, fixed, hard-to-palpate left laterocervical lymphadenomegaly; and the thoracic examination was negative.

Neck CT showed an inhomogeneous node at the third level of the neck on the left side, with a peripheral enhancement and a colliquative-necrotic center (Figure 2).

Echo-guided percutaneous needle biopsy was performed in April 2023 with a lymph node biopsy. The polymerase chain reaction (PCR) (multiplex real-time PCR essay) for M. tuberculosis was positive for two samples (36.18 and 36.74 cycle thresholds) and the histologic pattern denoted a giant cellular granulomatous necrotizing flogosis with giant cells mixed with epithelioid cells, lymphocytes, and plasma cells, a pattern that is compatible with tuberculosis (Figure 3).

Microscopic acid fast bacilli (AFB) tests and molecular examinations (real-time PCR) on induced sputum and bronchoalveolar lavage (BAL) indicated that the patient tested negative for mycobacterial cultures. Fiberoptic bronchoscopy showed just some whitish secretions in trachea, with a normal anatomic pattern in the right and left bronchial system. In BAL, the CD4/CD8 ratio was 3.3.

After the diagnosis of extrapulmonary TB (nodal), TB (microbiological criteria), and pulmonary TB (radiological criteria), the patient started standard oral antituberculosis therapy (rifampicine—600 mg/day, isoniazide—300 mg/day, ethambutol—1 g/day, and pyrazinamide—1 g/day) since she was sensible with the resistance genotypic test for first-line treatment. She discontinued omalizumab, continued with systemic steroids (oral prednisone 37.5 mg/day, tapered weekly to 6 mg/day), and was suspended in July 2023, with adequate CSU control after tapering steroids. The patient did not manifest any adverse reaction to antitubercular drugs and she did not show any organic symptoms connected to antituberculosis therapy, and a blood test showed that her liver function was normal after two months of therapy. The culture for *M. tuberculosis* on lymph node biopsy performed in April 2023 tested negative. Lymphonodal swelling disappeared after two months of antitubercolous therapy. The patient shifted to two antitubercular drug regimens (rifampicine (600 mg/day) and isoniazide (300 mg/day)) in July 2023. Neck CT, performed in September 2023 and compared with the previous CT in April 2023, confirmed, even if without a contrast medium, a dimensional reduction in left submandibular lymphonodes in the IIA and IIIA stations. In October 2023, the patient underwent microscopic acid fast bacilli (AFB) tests and molecular examinations (real-time PCR) on induced sputum, and they were again negative (mycobacterial culture is in progress). The lymph node follow-up was clinical and radiological, and the biopsy was not repeated. The patient suspended antitubercular therapy in October 2023.

## 3. Discussion

To our knowledge, this is the second case of TB diagnosis during omalizumab treatment, which suggests that attention ought to be paid to the known risk of TB during biotechnological treatments [17].

The patient was also treated with systemic steroids for CSU in the 2 weeks before being diagnosed with TB, but the duration of treatment should not be sufficient for TB activation [14].

Biologic agents are an expanding class of medications, accounting for more than 20% of all drugs approved annually by the U.S. Food and Drug Administration (FDA) since 2014 [18].

Monoclonal antibodies constitute the majority of approved biologic agents, and several options are now available for the management of allergic and immunologic disorders, including CSU. Although monoclonal antibodies have demonstrated efficacy and transformed clinical care for these multiple conditions, several factors must be considered when initiating and monitoring therapy. In this clinical decision process, risks of biologic therapy need to be understood in order to adequately counsel patients and appropriately monitor potential adverse events [19].

It is difficult to make conclusions about the risk of parasitic infections in patients receiving omalizumab due to insufficient data [20]. One double-blind placebo-controlled study assessed the risk of helminth infection among individuals treated with omalizumab who faced a high risk of helminth infection. All individuals received antihelmintic treatment, followed by 52 weeks of omalizumab or placebo treatment [13]. Those receiving omalizumab did not have a significantly higher risk of helminth infection (OR 1.47, 95% CI 0.74–2.95) [13]. A recent review reports that there is no evidence to suggest that omalizumab leads to increased susceptibility of other types of infection, and studies have not demonstrated an increased risk of opportunistic or serious infection [21].

According to current guidelines, tuberculosis infection screening is mandatory before starting any other kind of biotechnological therapy, such as anti-TNF-a or anti-IL12/23 [22,23,24].

Although it has been suggested that TB has a negative effect on the prevalence of allergic diseases [25], the number of studies in the literature on TB and such diseases is limited [26].

Diagnosis and treatment approaches to TB infection are changing rapidly. This welcomed development should allow for more aggressive and effective public health measures to be instituted by national TB control programs in order to speed progress towards TB elimination and consequently extend TB screening [5].

## 4. Conclusions

Even if current guidelines do not recommend screening for TB before starting anti-IgE therapy, further data should be sought to study the possible relationship between omalizumab treatment and active TB. Our experience suggests that screening for TB should be carried out in patients from highly tuberculosis-endemic countries before starting omalizumab therapy.

## Figures and Tables

**Figure 1 antibiotics-12-01655-f001:**
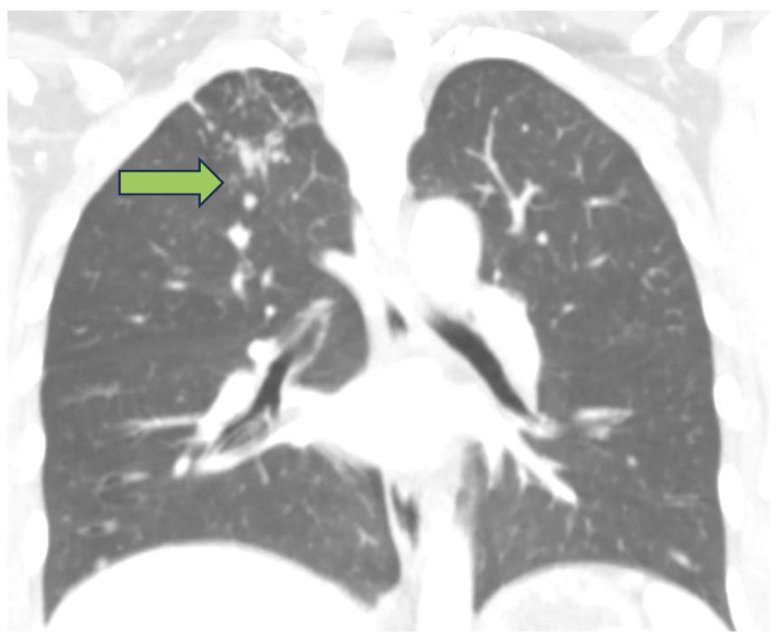
Computed tomography showing a “Tree in bud” (arrow) pattern at the upper-right pulmonary lobe ((**upper**): coronal-enhanced CT reconstruction, (**lower**): axial-enhanced CT reconstruction).

**Figure 2 antibiotics-12-01655-f002:**
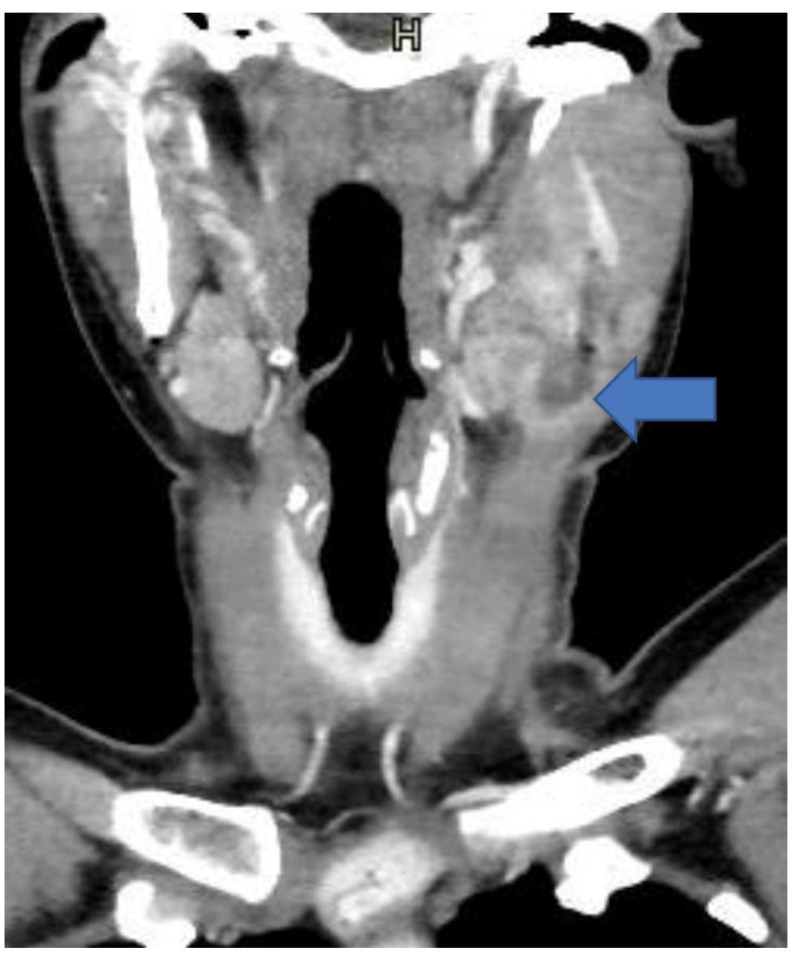
Computed tomography of the cervical region: Coronal-enhanced CT reconstruction shows a inhomogeneous node at the 3rd level of the neck on the left side, with a peripheral enhancement and a colliquative-necrotic center (arrow).

**Figure 3 antibiotics-12-01655-f003:**
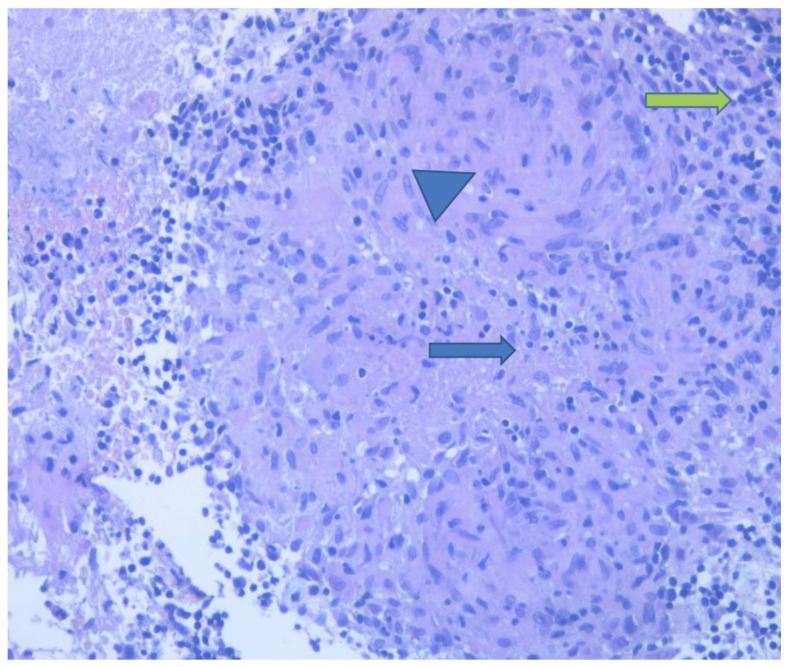
The pathological aspect of the biopsied later cervical lymphonode: granuloma with central eosinophilic necrosis (blue arrow) surrounded by epithelioid macrophages with pale eosinophilic cytoplasm (arrowhead) and peripheral lymphoctes (green arrow).

## Data Availability

Data will be available on request to corresponding authors.

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
