# Peer review of "Diagnosis of Tuberculosis in a Case of Chronic Urticaria following Omalizumab Therapy"

_antibiotics, 2023, doi:10.3390/antibiotics12121655_

Round 1

Reviewer 1 Report

Comments and Suggestions for Authors

I think this case study is interesting and helpful to our readers. However, the conclusions in the abstract and main text are overstated. I strongly recommend the conclusion needs to be modifed more humble.

Author Response

Thanks to the reviewer for the pertinent comment. We modified the conclusions more humble both in abstract and in main text:

"further data should be helpful are necessary to relate omalizumab treatment with active TB, and our experience may suggest high clinical TB suspicion in patients from highly tubercolous endemic countries if they are on omalizumab therapy and our experience may suggest screening for TB in patients from highly tubercolous endemic countries before starting omalizumab therapy.  " 

Please see also in main text 

Reviewer 2 Report

Comments and Suggestions for Authors

Dear Authors!

Hello, wherever you are!

I congratulate you on the article sent to the publisher!

The chosen topic is interesting, due to the practical importance of the biological antiallergic treatment performed in skin allergies.

After carefully reading the article, I send you the following comments and suggestions:

- the existence of editing flaws. In the body of the manuscript, a series of tables appear, some with text, which do not respect the editing rules of the publishing house.

-lack of full legends of the figures in the manuscript. The description of the figures must be made much more detailed, as it is also possible to indicate the types of histopathological lesions (with the help of several arrows). Also, a description of the nodal changes specific to nodal tuberculosis at the nodal ultrasound, visible changes on the ultrasound image inserted in the manuscript (e.g. ultrasonic signs of absent hilus, unclear edge, necrosis, echogenic thin layer, strong echoes, and capsular or peripheral vascularity);

- lack of conclusion;

- lack of a properly prepared Bibliography.

It is recommended to read the Author's Guide by the authors of the manuscript.

Good luck!

Author Response

The second reviewer wrote the following comments and suggestions to which we answered:   

1-lack of full legends of the figures in the manuscript. The description of the figures must be made much more detailed, as it is also possible to indicate the types of histopathological lesions (with the help of several arrows). Also, a description of the nodal changes specific to nodal tuberculosis at the nodal ultrasound, visible changes on the ultrasound image inserted in the manuscript (e.g. ultrasonic signs of absent hilus, unclear edge, necrosis, echogenic thin layer, strong echoes, and capsular or peripheral vascularity);

Thanks to the reviewer for the pertinent comment, we improved the description of the third picture and inserted more arrows, we substitute the second picture with a CT picture for better quality and we make a more detailed description: Figure 2: Coronal enhanced CT reconstruction  shows a inhomogeneous node in the 3rd level of the neck at the left side, with a peripheral enhancement and a colliquative-necrotic center (arrow).

Figure 3: Granuloma with central eosinophilic necrosis (blue arrow) surrounded by epithelioid macrophages with pale eosinophilic cytoplasm (arrowhead) and peripheral lymphoctes (green arrow). 

In the figures in text it is possibile to see graphic modifications

2- lack of conclusion;

Thanks to the reviewer for suggesting reading Author's Guide, we inserted conclusions:

Conclusions Even if current guidelines do not recommend screening for TB before starting anti-IgE therapy, further data should be helpful to study possible relation of omalizumab treatment with active TB, and our experience may suggest screening for TB in patients from highly tubercolous endemic countries before starting omalizumab therapy.

3 - lack of a properly prepared Bibliography.

As we wrote at beginning of cover letter for editor we have tried to improve bibliography adding some more citations and referring to more specific authors sentences in introduction and discussion

Reviewer 3 Report

Comments and Suggestions for Authors

In this manuscript, authors reported a case study by tracking a patient who suffered from chronic urticaria treated by topic steroid first, then by omalizumab therapy after several months, while this patient was not screened for tuberculosis (TB) infection. And five months later after applying omalizumab therapy, this patient was tested positive for TB by biopsy. Once the patient started standard antituberculosis therapy and discontinued omalizumab, the TB test showed negative, and lymph nodal swelling disappeared. By reading through it, I have some concerns:

1.      As for “Introduction” section, could authors reorganize the sentences, leading to several main paragraphs with a clear logic flow?

2.      As for “Case presentation” section, the reader can understand the key points easier if authors can provide a table to list the time point regarding the information including which therapy applied, what results demonstrated, and so on.

3.      Figure 1 shows the low resolution, and it may not highlight the points mentioned by authors, which is about the “tree in bud” pattern at upper right pulmonary lobe.

4.      In Figure 2, if authors can point out the key information on the image, that would be much clearer to understand the provided data. For example, why Figure 2 can show the positive for TB? How the control sample look like?

5.      It could be better to add the annotation of different types of cells to Figure 3. The current image in Figure 3 is lacking information to guide the reader.

6.      If possible, the negative result from lymph node biopsy could be compared to the data in Figure 2, to show the effect of different therapy at the same patient.

Thanks!

Author Response

The third reviewer wrote in particular the following comments and suggestions:  

  1. As for “Introduction” section, could authors reorganize the sentences, leading to several main paragraphs with a clear logic flow?
  2. As for “Case presentation” section, the reader can understand the key points easier if authors can provide a table to list the time point regarding the information including which therapy applied, what results demonstrated, and so on.

Thanks for suggestions, we tried to describe better the case, explaining the tests made and their outcomes, for example the colture of the only nodal biopsy was negative for Mycobacteria and we explicated that the biopsy was made in April 2023 when the patient come to our institute, it happens frequently to find DNA of Mycobacteria (with PCR) without growing of Mycobacterium and the clinical-radiologic follow up shows the correctness of diagnosis and usefulness of antitubercular therapy also in this case; we described also the update of follow up; 

  1. Figure 1 shows the low resolution, and it may not highlight the points mentioned by authors, which is about the “tree in bud” pattern at upper right pulmonary lobe.

Thanks for suggestion. We succeeded in obtaining a better resolution image of thorax (coronal and axial reconstruction)

  1. In Figure 2, if authors can point out the key information on the image, that would be much clearer to understand the provided data. For example, why Figure 2 can show the positive for TB? How the control sample look like?

Thanks for suggestion. We obtained a neck CT pictures and described it: 

Figure 2: Coronal enhanced CT reconstruction  shows a inhomogeneous node in the 3rd level of the neck at the left side, with a peripheral enhancement and a colliquative-necrotic center (arrow).

  1. It could be better to add the annotation of different types of cells to Figure 3. The current image in Figure 3 is lacking information to guide the reader.

Thanks a lot for suggestion, we annotate different cells in description and inserted more arrows in the picture: 

Figure 3: Granuloma with central eosinophilic necrosis (blue arrow) surrounded by epithelioid macrophages with pale eosinophilic cytoplasm (arrowhead) and peripheral lymphoctes (green arrow). Please see figures in main text

Reviewer 4 Report

Comments and Suggestions for Authors

This is interesting case report is on 32year old female suffering from chronic urticaria, treated with antihistamines, and short term (2-weeks) oral glucocorticoids. In November 2022 anti-IgE antibody (Omalizumab) was introduced. Five months later (April 2023) she was diagnosed with pulmonary tuberculosis and proper anti-tuberculosis treatment was successfully implemented. This was accompanied with discontinuation of Omalizumab therapy. Authors propose Omalizumab therapy as likely factor contributing to development of active tuberculosis infection, however, they don’t discuss possible effect of the previous therapeutic regimens that included oral corticosteroid. While the steroids are currently considered safe and beneficial for the treatment of several forms of tuberculosis (meningitis, pericarditis, miliary) there are anecdotal reports suggesting immune suppression with corticosteroids may predispose to tuberculosis. Authors should consider and discuss this possibility. Second query is on missing diagnosis of urticaria; omalizumab therapy strongly suggests allergic and IgE-mediated mechanism, however, no dermatologic tests are provided. It is also not clear how high IgE levels were before Omalizumab therapy; provided single value of >500 units during tuberculosis is moderately high and may reflect anti-IgE therapy rather than allergic/atopic reaction. 

Comments on the Quality of English Language

Minor misspelling noticed. 

Author Response

The fourth reviewer wrote in particular the following comments and suggestions:  

  • Authors propose Omalizumab therapy as likely factor contributing to development of active tuberculosis infection, however, they don’t discuss possible effect of the previous therapeutic regimens that included oral corticosteroid. While the steroids are currently considered safe and beneficial for the treatment of several forms of tuberculosis (meningitis, pericarditis, miliary) there are anecdotal reports suggesting immune suppression with corticosteroids may predispose to tuberculosis. Authors should consider and discuss this possibility.

Thanks a lot for this suggestion, we discussed this possibility inserting also one citation: 

The patient was also treated with systemic steroid for CSU for 2 weeks before diagnosis of TB but the duration of treatment should not be sufficient for TB activation [14].    

  • Second query is on missing diagnosis of urticaria; omalizumab therapy strongly suggests allergic and IgE-mediated mechanism, however, no dermatologic tests are provided. It is also not clear how high IgE levels were before Omalizumab therapy; provided single value of >500 units during tuberculosis is moderately high and may reflect anti-IgE therapy rather than allergic/atopic reaction. 

Thanks for this suggestion. Patient arrived to our infectious illnesses institute in April 2023 after referring to many specialists and many institutes, we tried to test her for basic tests about her skin problems, we reconstructed her clinical history before referring to us with the data she received from other specialist (therapeutic plan and so on); dermatologic iter she followed before coming to us  by the way arrived at prescription of omalizumab and we think it could be important to stress the necessity of analyzing possible causes of TB reactivation and spreading screening of TB both before immunologic therapies and in general.    

Round 2

Reviewer 2 Report

Comments and Suggestions for Authors

Dear Authors!

Please pay attention to some important aspects of the article.

1. In the body of the text it is not good to include tables with texts entered in these tables (Ex lines 36-40; 49-52; 59-63; 118-132)

2. Please correct the word ”day” from lines 121, 126, 127,

3. In Figure 1, it would be good to have arrows indicating the areas with the characteristic "tree in bud" appearance (top and bottom)

4. In the legend of Figure 2, it is indicated that there should be a title of the figure (Ex. Computer tomographic aspect of the cervical region. And then the explanation that you have presented correctly): row 161

5. In the legend of figure 3, it is indicated to have a title of the figure (Ex. The pathological aspect of the biopsied later cervical ganglion. And then everything you mentioned correctly in lines 166-168)

6. When mentioning bibliographic indexes (in the text), instead of [23], [24], [25], [23, 24, 25] will be mentioned: line 190

7. The word ”Conclusions” will become ”Conclusion”: line 197

8. In the Bibliography, it is good to have all the text and not a table (as you wrote)

9. The Bibliography must not contain: ””, italic font, vol, n, pp

10. For Bibliography 1, the citation rules of the electronic sites will be respected: World Health Organization. Global Tuberculosis Report 2021. Available on https://www.who.int. Accessed ... (date of access)

11. In the Bibliography, normally the first 6 authors are mandatory (if there are more than 6 authors), and only then is "et al" added.

12. In the Bibliography, the order of completing the names is strict: Name1, Surname1 (abbreviation); Name2, Surname2 (abbreviation);... (See MDPI and ACS Style): Example: Imperlini, E.; Massaro, F.; Buonocore, F. Antimicrobial Peptides against Bacterial Pathogens: Innovative Delivery Nanosystems for Pharmaceutical Applications. Antibiotics 2023, 12, 184. https://doi.org/10.3390/antibiotics12010184

Author Response

Thank a lot for the important suggestions. In this second round the second reviewer wrote in particular the following comments and suggestions:  

  1. In the body of the text it is not good to include tables with texts entered in these tables (Ex lines 36-40; 49-52; 59-63; 118-132)

We tried to solve this technical problem

  1. Please correct the word ”day” from lines 121, 126, 127,

We corrected this word as suggested

  1. In Figure 1, it would be good to have arrows indicating the areas with the characteristic "tree in bud" appearance (top and bottom)

We inserted arrow

  1. In the legend of Figure 2, it is indicated that there should be a title of the figure (Ex. Computer tomographic aspect of the cervical region. And then the explanation that you have presented correctly): row 161

We inserted the title as suggested

  1. In the legend of figure 3, it is indicated to have a title of the figure (Ex. The pathological aspect of the biopsied later cervical ganglion. And then everything you mentioned correctly in lines 166-168)

We inserted the title

  1. When mentioning bibliographic indexes (in the text), instead of [23], [24], [25], [23, 24, 25] will be mentioned: line 190

Using software for bibliography it is created automatically in this style but we insert a version without remote link (no tab)

  1. The word ”Conclusions” will become ”Conclusion”: line 197

We changed the word as suggested

  1. In the Bibliography, it is good to have all the text and not a table (as you wrote)

Using software for bibliography it is created automatically in this style but we insert a version without remote link (no tab)

  1. The Bibliography must not contain: ””, italic font, vol, n, pp

We corrected the bibliography as suggested

  1. For Bibliography 1, the citation rules of the electronic sites will be respected: World Health Organization. Global Tuberculosis Report 2021. Available on https://www.who.int. Accessed ... (date of access)

We corrected the bibliography as suggested

  1. In the Bibliography, normally the first 6 authors are mandatory (if there are more than 6 authors), and only then is "et al" added.

We corrected the bibliography as suggested

  1. In the Bibliography, the order of completing the names is strict: Name1, Surname1 (abbreviation); Name2, Surname2 (abbreviation);... (See MDPI and ACS Style): Example: Imperlini, E.; Massaro, F.; Buonocore, F. Antimicrobial Peptides against Bacterial Pathogens: Innovative Delivery Nanosystems for Pharmaceutical Applications. Antibiotics 2023, 12, 184. https://doi.org/10.3390/antibiotics12010184

We corrected the bibliography as suggested

See please the file with new text, figure legends and bibliography

Reviewer 3 Report

Comments and Suggestions for Authors

Thanks for the responses to my concerns. I think the updated manuscript has been revised well and good for publication. Please recheck the format through the revised one, such as:

Ln106-109: fonts style and size

Ln118-132: line spacing

Good luck : )

Author Response

Thanks very much for suggestions. The third reviewer wrote in particular the following comments and suggestions:  

  1. Please recheck the format through the revised one, such as:

Ln106-109: fonts style and size

Ln118-132: line spacing

We recheked fonts style and size

See please the file with new text, figure legends and bibliography

Reviewer 4 Report

Comments and Suggestions for Authors

Revised version of this interesting case addressed most if not all queries raised by initial review and improved quality of presentation significantly. This is the second report case of TB re/activation that may be linked to therapy with anti-IgE antibody (Omlaizumab) and thus it may be of high interest to clinical immunologists. 

Author Response

Thanks a lot for comment. Fourth reviewer wrote a comment about first resubmission which addressed most if not all queries raised by initial review and improved quality of presentation significantly.   

See please the new file with new text, figure legends and bibliography